# Parent, patient and clinician perceptions of outcomes during and following neonatal care: a systematic review of qualitative research

James Webbe,[1] Ginny Brunton,[2] Shohaib Ali,[3] Nicholas Longford,[1] Neena Modi,[1] Chris Gale,[1] the Core Outcomes in Neonatology (COIN) Project Steering Group

► Additional material is published online only. To view please visit the journal online (http://dx.doi.org/10.1136/bmjpo-2018-000343).

[1]Neonatal Medicine, Imperial College London, London, UK
[2]UCL Institute of Education, London, UK
[3]Imperial College London, London, UK

**Correspondence to**
Dr Chris Gale; christopher.gale@imperial.ac.uk

## ABSTRACT

**Objective** Multiple outcomes can be measured in infants that receive neonatal care. It is unknown whether outcomes of importance to parents and patients differ from those of health professionals. Our objective was to systematically map neonatal care outcomes discussed in qualitative research by patients, parents and healthcare professionals and test whether the frequency with which outcomes are discussed differs between groups.

**Design** Systematic review of qualitative literature. The following databases were searched: Medline, CINAHL, EMBASE, PsycINFO and ASSIA from 1997 to 2017. Publications describing qualitative data relating to neonatal care outcomes, reported by former patients, parents or healthcare professionals, were included. Narrative text was analysed and outcomes grouped thematically by organ system. Permutation testing was applied to assess an association between the outcomes identified and stakeholder group.

**Results** Sixty-two papers containing the views of over 4100 stakeholders were identified; 146 discrete outcomes were discussed; 58 outcomes related to organ systems and 88 to other more global domains. Permutation testing provides evidence that parents, former patients and health professionals reported outcomes with different frequencies (p=0.037).

**Conclusions** Parents, patients and health professionals focus on different outcomes when discussing their experience of neonatal care. A wide range of neonatal care outcomes are reported in qualitative research; many are global outcomes relating to the overall status of the infant. The views of former patients and parents should be taken into consideration when designing research; the development of a core outcomes set for neonatal research will facilitate this.

## INTRODUCTION

In high-resource settings approximately 1 in 10 babies will require care in a neonatal unit.[1] Conditions such as preterm birth affect patients' long-term outcomes: consequences include cardiovascular disease in adulthood,[2] neurosensory impairment,[3] respiratory disease[4] and lower rates of employment and

### What is already known on this topic?

► Multiple outcomes can be measured in infants that receive neonatal care.
► It is not known which outcomes are considered important by former neonatal patients, parents and healthcare professionals, or whether these differ between groups.

### What this study hopes to add?

► The predominant outcomes identified by parents, former patients and health professionals related to holistic concepts (such as 'normality').
► Significant differences were identified in outcomes discussed by parents, patients and health professionals.
► Differences in neonatal outcomes prioritised by parents, patients and health professionals should be recognised when planning research.

marriage.[5] Infants born more prematurely tend to have worse outcomes.[6] As neonatal survival for babies of all gestational ages improves long-term outcomes become more important.

An outcome is the measured effect that illness or treatment has on an individual.[7] Parents and patients are rarely involved in outcome selection in paediatric research.[8] Poor outcome selection causes research waste[9]: research produced is not relevant to patients' lives. Neonatal care, and the underpinning research, should focus on outcomes important to those it affects most: former neonatal patients, parents and healthcare professionals.[9 10] Identifying these outcomes is crucial to ensure research is relevant and efficient.[9 11] Qualitative research provides a rich description of complex phenomena such as neonatal care.[12] One commonly used approach to identify outcomes of importance

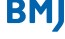

to stakeholders is primary qualitative research. Considerable qualitative research exploring how parents and health professionals perceive neonatal care has been conducted previously[13][14]; therefore, by systematically reviewing published qualitative research it is possible to map the outcomes discussed by different groups. This review does not include all research on how stakeholders perceive neonatal care: it is focused on how former patients, parents and health professionals perceive the outcomes of this care.

In this study we aimed to map the range of outcomes identified in qualitative literature by different stakeholder groups: parents, ex-neonatal patients and healthcare professionals. We also wanted to test the hypotheses that stakeholder groups prioritise outcomes differently, and that outcomes identified differ by infant gestational age category.

This work is a component of a wider programme to compile a core outcomes set for neonatology.[15] A core outcomes set is an agreed collection of important outcomes identified through robust consensus methods by all key stakeholder groups.[7] The results of this study will be combined with the results of a systematic review of outcomes reported in clinical trials.[16] These will be used as the starting point for the consensus process to determine a core outcomes set.[15]

## METHODS

We registered this systematic review prospectively on PROSPERO (prospective register of systematic reviews): CRD42016037874.[17] We conducted the review according to Preferred Reporting Items for Systematic Reviews and Meta-Analyses (PRISMA) guidelines.[18] We searched the following databases: Medical Literature Analysis and Retrieval System Online (MEDLINE), Cumulative Index to Nursing and Allied Health Literature (CINAHL), Excerpta Medica Database (EMBASE), Psychological Information Database (PsycINFO) and Applied Social Sciences Index and Abstracts (ASSIA). Qualitative or mixed methods studies were included if they contained outcomes identified by stakeholders in the context of babies admitted to a neonatal unit. Full inclusion and exclusion criteria are listed in online supplementary eTable 1. We considered all studies published from 1 January 1997 to 1 January 2017 in a peer review journal in all languages (where necessary a translation was obtained). The databases were last searched on 14 February 2017. The search strategy used for MEDLINE is described in online supplementary eFigure 1. The terms derived from this search strategy were translated to other databases.

All identified papers were screened by title and abstract and then by full text. After double-screening a sample of papers and agreeing criteria all screening was completed by one researcher (JW). For quality assurance, a second researcher screened a random 10% sample of abstracts

> **Box 1  An example of an outcome hierarchy**
>
> ► Text extracts to identify or infer a result of clinical care, the 'outcome' such as '*Bonding with parents*'.
> ► Similar 'outcomes' were grouped into thematically linked 'domains' such as '*Relationships with others*'.
> ► 'Outcome domains' relating to similar concepts were grouped into 'categories' such as '*Social*'.
> ► We did not address the ways in which an outcome was measured. For example, the 'outcome' '*Parental bonding*' could be measured using parent-reported scores or an external assessment.

and titles (CG). Agreement between reviewers was assessed by Cohen's kappa coefficient.[19]

After screening all papers were coded independently by two researchers (JW and CG or GB) using Eppi-Reviewer V.4 software.[20] Any disagreement was resolved by a third researcher (CG or GB). Data on study design, stakeholder demographics, infant birth characteristics and verbatim text relating to neonatal care outcomes were extracted and stored. Quality assessment of individual studies was not undertaken as it is a controversial area of uncertain value in relation to qualitative research.[21]

All outcomes were grouped according to a previously defined framework of organ systems[22] using the following domains: cardiovascular, respiratory, gastrointestinal, neurological, genitourinary, infection, skin and development. All three reviewers jointly refined this framework using methods incorporating thematic analysis.[23–25] Where narrative data did not fit clearly into the domains, dialogue between all reviewers was used to develop new domains. Outcome domains were thematically analysed to develop higher order categories. A new hierarchy was developed to group outcomes because established hierarchies either did not relate well to neonatal care[26–28] or missed key concepts.[7] This outcome hierarchy is described in box 1.

We analysed whether outcomes identified differed by stakeholder groups and by infant gestational age category (using WHO definitions of prematurity).[29] We used permutation testing[30] to test for an association between the frequency that outcomes in different domains were identified and the stakeholder group involved. We performed 5000 replications to generate the distribution of the test statistic under the null hypothesis of no association, and compared our results with this distribution. We performed a similar analysis to test for an association between infant gestational age and frequency of outcome reporting. If a significant association was found we explored this further in a post hoc analysis to identify where the observed results differed most from the frequencies expected under the hypothesis of no association established by the permutation analysis.

## RESULTS

Database searches produced 1130 results which were screened and assessed for eligibility (figure 1). After

applying inclusion and exclusion criteria 62 studies containing the views of 4100 stakeholders were analysed. Agreement between reviewers was high (Cohen's kappa coefficient=0.81).[19]

The 62 included studies reported data from 15 countries; 9 related to full-term infants, 31 to preterm infants (born <37 weeks' gestational age) and 20 to extremely preterm infants (born <28 weeks' gestational age). A range of methodologies was used including direct observation (13 studies) and individual (25 studies) or group interviews (13 studies). Questionnaires were used in 21 studies, two of which were Delphi processes. Included studies are described in online supplementary eTable 2.

Included studies involved over 4100 participants. Parents were the most frequently involved stakeholder group (1969 parents in 40 studies; 65%); former neonatal patients were less commonly included (368 patients in 5 studies; 8%). Nurses and midwives were the professional group involved most often (1096 involved in 24 studies; 39%). Three hundred and sixteen doctors were involved in 18 studies (29%). We also identified 351 additional participants consisting of other family members, teachers, social workers and allied health professionals. In many studies, particularly those employing observation of clinical practice, the total number of research participants was not recorded.

One hundred and forty-six distinct outcomes were extracted from the included studies. Fifty-eight outcomes related to organ systems within the original framework; we were unable to categorise 88 outcomes within the

original framework. The final framework is shown in table 1. An example of the thematic analysis leading to the expanded framework is shown in box 2.

The full inventory of outcomes is listed in online supplementary eTable 3. A table of all outcomes in each study (with verbatim text extracts) is shown in online supplementary eTable 4.

Outcomes were identified relating to all of the organ systems included in the original framework and assigned to an organ system outcome domain category (table 2). The organ system outcome domains most frequently discussed at the study level were 'development' (32 studies; 52%) and 'gastrointestinal' (24 studies, 39%). The individual organ system outcomes most frequently discussed were 'language disorders' (8 studies, 13%), 'visual impairment' (7 studies, 11%) and 'breast feeding' (7 studies, 11%).

The majority of outcomes identified did not relate to individual organ systems. Some related to the overall status of the infant and were assigned to a holistic outcome domain category (table 3). Other domains related to the effects experiencing neonatal care has on parents; these were assigned to a 'Parent focused' outcome domain category (table 4). Another group of domains related to the neonatal care delivered; these were assigned to a 'Healthcare delivery' outcome domain category (table 5). A group of domains was identified relating to the cost of neonatal care; these were assigned to an 'Economic' outcome category (table 6). Finally, a group of outcome domains was identified relating to the relationships neonatal patients develop with others;

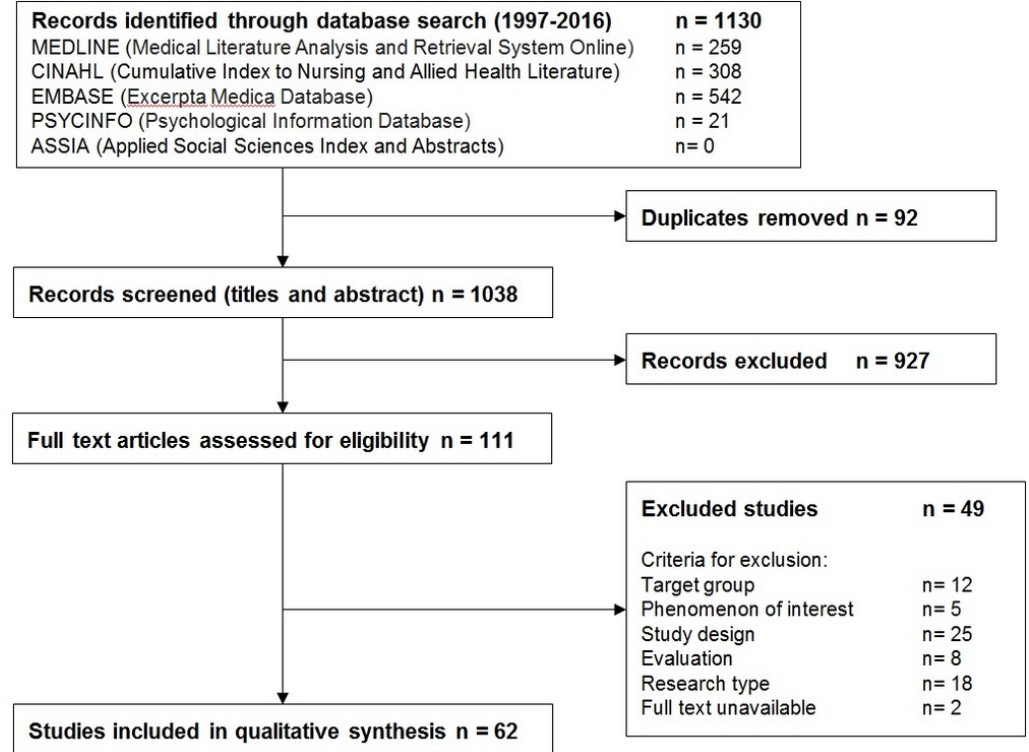

**Figure 1** PRISMA flowchart of study selection.

**Table 1** Final outcome framework

| Outcome domain categories | Outcome domains |
|---|---|
| Organ system outcomes | Cardiovascular |
| | Respiratory |
| | Gastrointestinal |
| | Neurological |
| | Genitourinary |
| | Infection |
| | Skin |
| | Developmental |
| *Holistic outcomes* | *Survival* |
| | *Growth* |
| | *Pain* |
| | *Suffering* |
| | *Normality* |
| | *Other outcomes* |
| *Parent-focused outcomes* | *Parental support* |
| | *Other outcomes* |
| *Healthcare delivery outcomes* | *Healthcare workers—knowledge and competence* |
| | *Healthcare workers—Communication* |
| | *Other outcomes* |
| *Economic outcomes* | *Healthcare utilisation* |
| | *Other outcomes* |
| *Social outcomes* | *Psychiatric outcomes* |
| | *Relationships with others* |
| | *Other outcomes* |

Outcome domain categories and outcome domains added to the original framework marked in *italics*.

**Box 2  Example of framework synthesis related to the outcome of 'Normality'. Thematic analysis of verbatim extracts identified a recurring theme**

► *'The mother also worried that…Lisa would not have a normal life.'*[41]
► *'Being reassured that he was on line for how old he was…Just reassurance he was doing well.'*[42]
► *'Finally, a mother called it a developmental land-mark when an older sister dared show her irritation towards her little brother, 'no longer treating him as if he were made of glass.'*[43]
► From this and similar text the outcome of 'Normality' was derived by thematic analysis. It did not fit within the existing outcome hierarchy but was reported extensively, so a new domain was added to the framework again called 'Normality'. This outcome domain relating to the overall status of the infant was similar to outcome domains like 'survival', 'vitality' and 'growth', so these domains were grouped together as an outcome domain category called: 'Holistic outcomes'.

## DISCUSSION

We have systematically reviewed and synthesised the outcomes reported in qualitative research by those with lived experience of neonatal care: patients, parents and healthcare professionals. We show that the patterns of outcomes discussed by former neonatal patients, parents and healthcare professionals are different. This is in keeping with previous single-centre research[31] and case reports.[32] This indicates that healthcare professionals should consider whether the outcomes they discuss align with patients and parents' concerns.[33] Acceptance of the differences shown should form part of the process of shared decision-making in clinical practice.[34] Poor outcome selection is also a known problem in paediatric research,[8 35] involving patients and parents will help reduce research waste.[36 37]

The outcomes identified extend beyond the organ system-specific outcomes commonly reported in clinical trials and include global concepts such as 'normality' of the child in later life, the impact on an infant's family and the healthcare team, financial and time costs and how patients interact with wider society. Our findings are in keeping with observational studies illustrating the wide-reaching consequences of neonatal illness.[38–40] Another feature of the outcomes identified is that rather than relating to a specific diagnosis or disease many reflect the global status of the child. Diagnoses like necrotising enterocolitis or retinopathy of prematurity were mentioned less frequently than their consequences, such as feeding difficulties or visual impairment. In general, the outcomes identified indicate that pathological processes and diagnoses are less relevant to patients and parents than the effects they have on day-to-day life. Priority should be given to identifying efficient ways of measuring more global outcomes of neonatal conditions throughout childhood and later life, for example, through robust linkage of neonatal data with education databases.

these were assigned to a 'Social' outcome domain category (table 7).

From these outcome domains the most frequently discussed at study level were 'parental support' (30 studies, 48%) and 'healthcare workers—communication' (30 studies, 48%). The individual outcomes most frequently discussed were 'normality' (16 studies, 26%) and 'survival' (11 studies, 18%).

Permutation test analysis showed a statistically significant association (p=0.037) between different stakeholder groups and outcome categories identified (online supplementary eFigure 3). The frequency with which patients discussed the outcomes was most divergent from the other groups. In particular, patients discussed outcomes relating to the genitourinary, surgical, developmental and pain outcome domains more than would be expected by chance.

We found no statistically significant association (p=0.114) between gestational age and outcome categories (online supplementary eFigure 2).

**Table 2** Organ system outcomes

| Organ system outcome domains | Number of studies discussing outcome domain (n=62) | Outcome | Number of studies discussing outcome (n=62) | Verbatim text extract |
|---|---|---|---|---|
| Developmental | 32 | Ability to walk | 3 | 'He walked four, my mother never forgot, she says it was a miracle of God.'[44] |
| | | Difficulties with activities of daily living | 4 | 'A lot of times I have to put myself in certain positions to do things, like opening a jar—I have to squeeze it in between my arms instead of gripping it with my hand.'[45] |
| | | Hearing impairment | 5 | 'I told the parents that he will never be able to see, to hear and I will get more data to show them how bad things are.'[46] |
| | | Issues of development and motor skills | 5 | 'We both looked at my child, research, experience and what I could expect.'[47] |
| | | Language disorders | 8 | 'I also had a hard time learning to talk.'[48] |
| | | Social difficulties | 2 | Social communication is difficult because of his hearing and speech problems, and he is described as having a few friends and no experience in dating.[49] |
| | | Visual impairment | 7 | At the time of the interviews, the only major sequel was in one child with seriously impaired vision.[43] |
| | | Other outcomes only in 1 paper | | Ability to feed themselves: ability to undertake sport: need for physical therapy: normal hearing: retinopathy of prematurity |

Continued

**Table 2** Continued

| Organ system outcome domains | Number of studies discussing outcome domain (n=62) | Outcome | Number of studies discussing outcome (n=62) | Verbatim text extract |
|---|---|---|---|---|
| Gastrointestinal | 24 | Breast feeding | 7 | 'I fully breastfed for 4 months—100%—and I am so proud of it.'[47] |
| | | Choice of milk for feeding | 2 | 'It's like they [scn providers] didn't inform us when they were trying to feed my daughter [formula].'[50] |
| | | Feeding difficulties | 5 | 'We kept on saying to the staff on neonatal unit that it was only Gray's feeding that was stopping him from going home, everything else was fine.'[51] |
| | | Feeding practices | 2 | Mothers had difficulty understanding these instructions and seemed hesitant to liberalise their infant's intake after discharge.[52] |
| | | Initiating enteral feeds | 2 | 'MEF [minimal enteral feeds] should be initiated in first 2–3 days of life as long as the baby is stable.'[53] |
| | | Oral feeding | 3 | '[The] very first time [feeding the baby] was just great, to tell you the truth.'[54] |
| | | Other outcomes only in 1 paper | | Choking during feeding: eating disorder: fistulas: frequency of defecation: liver failure: necrotising enterocolitis: nutritional intake: other gastrointestinal malformations: regurgitation: short gut syndrome |
| Respiratory | 12 | Frequent respiratory illnesses | 2 | 'There were lots of masks and nebulisers during those years.'[43] |
| | | Mechanical ventilation | 5 | Over 30% of all infant descriptions were about babies who had tracheostomies and were unable to be weaned off a ventilator.[55] |
| | | Oxygen dependence | 5 | 'My babies did not fit into the criteria for going home early due to one of the twins still being dependent on oxygen.'[51] |
| | | Other outcomes only in 1 paper | | Asthma: breathlessness: chronic lung disease: excessive secretions: nasal congestion: pneumothorax |

Continued

**Table 2** Continued

| Organ system outcome domains | Number of studies discussing outcome domain (n=62) | Outcome | Number of studies discussing outcome (n=62) | Verbatim text extract |
|---|---|---|---|---|
| Neurological | 11 | Brain damage (not further specified) | 2 | 'Brain injury is laden with a lot more emotions and moral concerns for sure.'[56] |
| | | Neurological symptoms | 2 | 'Can't feel some—my left—this is numb right here.'[57] |
| | | Seizures | 2 | 'I explained this to the doctor. And he was the one that said it could possibly be seizures. So we're going to take him in and have him tested.[58] |
| | | Significant IVH | 2 | 'Although she has a grade IV bleed, the resident says that she moves and looks around, and he thinks the odds are quite good.[46] |
| | | Sleep disorders | 4 | Subsequent to an account of the son's disturbed sleep at night for several months after discharge, which was an enormous challenge to the couple.[43] |
| | | Other outcomes only in 1 paper | | Neurological care |
| Surgical | 5 | Appearance of scars | 2 | "I do not like the scar on my belly [...]; I was at the beach and everyone kept staring at me like 'That is a big scar'."[48] |
| | | Need for multiple operations | 2 | The mother also worried that there would be more surgeries.[41] |
| | | Other outcomes only in 1 paper | | Care for surgical babies: need for ileostomy |
| Infection | 5 | Sepsis | 3 | Decrease bloodstream infections selected as key performance indicator[59] |
| | | Other outcomes only in 1 paper | | Prevention of infection: susceptibility to infection |
| Skin | 4 | Appearance of scars | 2 | In addition, hospitalisation and different interventions in their first days of life have left marks on their bodies.[48] |
| | | Other outcomes only in 1 paper | | Burns: extravasation injuries: pressure sores: skin care |
| Cardiovascular | 1 | Other outcomes only in 1 paper | | Hypotension: presence of patent ductus arteriosus |
| Genitourinary | 1 | Other outcomes only in 1 paper | | Urological disorders |

IVH, intraventricular haemorrhage.

**Table 3** Holistic outcomes

| Holistic outcome domains | Number of studies discussing outcome domain (n=62) | Outcome | Number of studies discussing outcome (n=62) | Verbatim text extract |
|---|---|---|---|---|
| Normality | 22 | Ability to lead a normal life | 2 | 'The mother also worried that… Lisa would not have a normal life.'[41] |
| | | Normality | 16 | "A major focus for parents was seeking information that told them that what was happening was 'normal' and that everything was going to be 'fine'."[60] |
| | | Other outcomes only in 1 paper | | Being treated normally: inability to create a normal life: normal health: thriving |
| Suffering | 15 | Comfort | 4 | 'You can almost feel what it's like in the incubator, lying on the lambskin, that it's how I would want to have laid and… Well, it looks very comfortable.'[61] |
| | | Suffering | 9 | 'This infant's short life was never comfortable…I frequently felt we were torturing the child just doing daily care.'[55] |
| | | Other outcomes only in 1 paper | | Ex-patients' separation from their suffering: symptom control |
| Survival | 14 | Survival | 11 | 'It hurts. I didn't know, I didn't know if they were going to make it or not.'[58] |
| | | Survival with disability | 3 | 'It isn't up to us to say what is quality of life, because parents might think that even if the child was disabled, that it was better than not having a child at all.'[62] |
| | | Survival without disability | 4 | 'And afterwards you are worried about how they are going to survive. If they would have impairments, and so on.'[43] |
| Growth | 8 | Growth | 8 | 'She was born so early, it's nice to see that she's finally catching up with how she's growing.'[42] |
| Pain | 7 | Pain | 4 | 'It like hurts when you wake up in the morning.'[57] |
| | | Pain management | 2 | Research priorities identified: identifying effective interventions to prevent or reduce pain or stress[63] |
| | | Other outcomes only in 1 paper | | Chronic pain |

Continued

**Table 3** Continued

| Holistic outcome domains | Number of studies discussing outcome domain (n=62) | Outcome | Number of studies discussing outcome (n=62) | Verbatim text extract |
|---|---|---|---|---|
| Other outcomes | | Overall health state | 2 | 'We try to think of the whole life consequence. We talk about, you know, strength and cognitive capacity, but also life and communication and feeding yourself and getting around.'[56] |
| | | Vitality | 2 | 'The doctor said that, whatever we do, however good we are, it is mostly up to the infant himself. No matter how small they are, they can have something within themselves.'[64] |
| | | Physical appearance | 7 | Both mothers and fathers found their infant's appearance and behaviour to be the stressors with the most impact.[65] |
| | | Other outcomes only in 1 paper | | Physiological stability |

This more holistic approach should extend to how babies are categorised. Our work included an undoubtedly heterogeneous population, but this was driven by discussions with former neonatal patients and parents at the planning stages of this project. They strongly stated that 'a sick baby is a sick baby' regardless of birth weight or gestational age: a statement that is supported by our finding that there was no significant difference in how frequently outcomes were discussed in relation to babies of differing gestational ages. Splitting research populations by arbitrary landmarks not recognised by parents or former patients[32] may be a source of research heterogeneity.

The strengths of our study included identification and synthesis of outcomes from an international and methodologically diverse range of studies, relating to babies of all gestational ages, and a wide range of stakeholders. We included outcomes that stakeholders spontaneously identified. As a result, we were able to include data from a wider range and diversity of stakeholders than a primary research study could. We followed a preregistered protocol with reporting in line with PRISMA guidelines.[18] It has been argued that quality assessment is needed in 'mapping' reviews to aid in interpretation and uptake of findings,[24] but the value of this approach is uncertain.[21] The consultation phase of our core outcomes set development work will provide the opportunity to critically reflect on the contribution of these findings to our understanding of what constitutes an 'important' outcome in neonatal research.

A limitation of our study is that, in line with many systematic reviews, we are synthesising data from studies that did not explicitly address the research question we are asking. This meant that we combined data about which outcomes parents, patients or healthcare professionals mentioned during research. As a result, we described how frequently outcomes were discussed, rather than the importance assigned by groups to each outcome. Many outcomes were only discussed in a single study. We present them here to show the range and breadth of outcomes discussed, but cannot comment on whether they are more or less important than more frequently mentioned outcomes. Another limitation is that the researchers who undertook the primary qualitative research in the included studies will have influenced our review through their analysis; we reviewed data that was a step removed from the opinions of the stakeholders themselves. However, by following rigorous methodology and employing a comprehensive search strategy we have combined all available data to produce this mapping review.

Trying to measure all of the varied outcomes identified in this work in research would be impractical, if not impossible. This work supports the importance of identifying a core outcomes set, and highlights the importance of input from all stakeholder groups. In other fields, core outcomes sets have successfully aligned patient and healthcare professional research priorities.[36]

**Table 4** Parent-focused outcomes

| Parent-focused outcome domains | Number of studies discussing outcome domain (n=62) | Outcome | Number of studies discussing outcome (n=62) | Verbatim text extract |
|---|---|---|---|---|
| Parental support | 30 | Coping with maternal illness | 5 | One nurse spoke of her belief that mothers could be diagnosed with depressive conditions.[62] |
| | | Culture differences | 2 | Three families felt strongly that their stress derived from differences in the medical management approaches between the USA and their homeland.[65] |
| | | Parental ability to work | 2 | They liked being back at work because it helped occupy their minds, but they reported being exhausted.[66] |
| | | Parental competence | 4 | 'We learned everything we needed and knew what we had to do, I was quite comfortable when we went home.'[47] |
| | | Parental involvement | 10 | 'During our baby's stay in the neonatal unit both myself and Peter were fully involved in our son's care and did most of the caring such as nappy changing and NGT feeds.[51] |
| | | Support from family and friends | 5 | 'My mother in law and my mother both would watch my older daughter that first year quite a bit while I would take my daughter to therapy.[45] |
| | | Support from fathers | 2 | "Fathers ranged from being very supportive, '[we're] in this together,' to being deterrents or completely absent."[66] |
| | | Support from healthcare professionals | 6 | 'The nursing staff, the doctors…they really know what they're doing…not only medically, but dealing with us personally… that helped a lot.[67] |
| | | Other outcomes only in 1 paper | | Balancing caring for themselves and their baby: barriers to parental involvement: care provided close to home: maintaining hope: online support: preparation for NICU admission: support from faith |
| Other outcomes | | Long-term effects on parents | 2 | We should be looking at Postnatal Depression after the baby goes home… Once they actually get a baby home, that's when the reality sets in. All the triggers are there.[62] |
| | | Other outcomes only in 1 paper | | Support beyond NICU: parental perception of uncertainty |

NGT, nasogastric tube; NICU, neonatal intensive care unit.

**Table 5** Healthcare delivery outcomes

| Healthcare delivery outcome domains | Number of studies discussing outcome domain (n=62) | Outcome | Number of studies discussing outcome (n=62) | Verbatim text extract |
|---|---|---|---|---|
| Healthcare workers—communication | 30 | Communicating in challenging settings | 10 | When they arrived at hospital…some had difficult conversations with clinical staff… NICUs commonly set boundaries around the care that they offer.[68] |
| | | Communicating information effectively | 7 | Other parents experienced stress from unknown medical terminology.[65] |
| | | Communication about discharge | 3 | Parents/caregivers may be inadequately prepared for home care and management of fragile neonates due to a lack of consistent and early communication.[69] |
| | | Communication with parents | 2 | 'When you're talking to parents while you're doing cares and everything, you're not really talking to them.… you're having a vague conversation across the room.'[70] |
| | | Developing a caring relationship | 5 | As the providers gave support to families, families also were described as supporting the staff.[55] |
| | | Keeping parents informed | 7 | 'I asked so many questions and read all the charts every day, and I probably angered them. Squeaky wheel gets the oil, as they say.'[50] |
| | | Treating parents with respect | 3 | 'I got yelled at by a nurse at [the scn] for rubbing my son's foot [even though that was okay at the nicu].'[50] |
| | | Other outcomes only in 1 paper | | Allowing time for conversation: awareness of parental views: candour: communication with ex-neonatal patients: language barrier: using aids to communication |

Continued

**Table 5** Continued

| Healthcare delivery outcome domains | Number of studies discussing outcome domain (n=62) | Outcome | Number of studies discussing outcome (n=62) | Verbatim text extract |
|---|---|---|---|---|
| Healthcare workers—knowledge and competence | 23 | Consistency of decisions | 6 | 'Everybody had a different point of view but they were opinions, not facts. So that was huge, don't even get me started on that, that was just a nightmare.'[60] |
| | | Ethical decision-making | 5 | "…but when you actually mention this, say, 'Well, in fact you know, we don't really know what's the best treatment,' it is a delicate moment."[71] |
| | | Healthcare professionals' behaviour | 5 | 'It wasn't a nurse related conversation it was just a casual conversation… Like I felt a bit [sic] she wasn't their priority.'[60] |
| | | Healthcare professional competence | 7 | Most of the parents recalled specific incidents that they perceived as poor medical care; typically, these incidents involved technical procedures or medical knowledge.[72] |
| | | Identifying who is responsible for care | 3 | 'Sometimes we're not real clear who to follow up with.'[50] |
| | | Staffing levels | 2 | It was especially helpful for the parents to see so many nurses and physicians in the NICU.[73] |
| | | Other outcomes only in 1 paper | | Expertise in palliative care: medical errors: staff insecurity |
| Other outcomes | | Iatrogenic harm | 3 | 'There are potential toxicities, very real toxicities associated with it.'[71] |
| | | Inclusion in research | 2 | Parents were often interested in the research, and some would have liked more contact and information than they actually received.[68] |

NICU, neonatal intensive care unit.

**Table 6** Economic outcomes

| Economic outcome domains | Number of studies discussing outcome domain (n=62) | Outcome | Number of studies discussing outcome (n=62) | Verbatim text extract |
|---|---|---|---|---|
| Healthcare utilisation | 15 | Frequent appointments | 2 | *'I felt left out, I was always missing school because I had to go to the hospital for check-ups.'[48]* |
| | | Frequent readmissions | 4 | *The prolonged hospitalisations experienced by children with BPD and the frequent interactions of families with medical personnel may result in increased access and opportunities for services for parents of children with BPD.[74]* |
| | | Inappropriate treatments | 2 | *Community providers…may lack the required knowledge and skills to manage complex infants, leading to suboptimal office-based care and perceived overutilisation of the emergency system.[69]* |
| | | Need for frequent treatments | 3 | *'There were lots of masks and nebulisers during those years.'[43]* |
| | | Need for lifelong care | 3 | *'When the outcome is disastrous they just expect parents to take home severely handicapped babies and deal with life-long problems.'[75]* |
| | | Recurrent sickness | 1 | *'We've only put him with other children for the past month. The biggest worry right now is when he is going to get sick.'[58]* |
| Other outcomes | | Duration of admission | 2 | *Decrease length of stay selected as key performance indicator[59]* |
| | | Healthcare resources | 3 | *Although respondents frequently discussed the emotional toll to all concerned, the monetary cost of long-term stays was very rarely (<1%) mentioned.[55]* |

BPD, borderline personality disorder.

**Table 7** Social outcomes

| Social outcome domains | Number of studies discussing outcome domain (n=62) | Outcome | Number of studies discussing outcome (n=62) | Verbatim text extract |
|---|---|---|---|---|
| Relationships with others | 19 | Bonding with family and friends | 3 | *'The only thing we might have done…some of our closest friends…it would have been nice to have them there as well.'[67]* |
| | | Bonding with parents | 8 | *'I find it a great joy when the mums do hold the baby against their chest.'[76]* |
| | | Effects on family and friends | 7 | *Almost all parents acknowledged the emotional adjustment of other family members in response to raising a child with physical impairment.[45]* |
| | | Family resources | 2 | *Three families felt overwhelmed by a lack of resources (especially in the area of family support).[65]* |
| | | Peer acceptance | 2 | *I've had 4 year-olds tell me the other kids don't want to play with them cause they have a dumb arm.[57]* |
| | | Other outcomes only in 1 paper | | Childhood happiness: overprotective parent–child relationship: psychological coping |
| Psychiatric | 7 | Need for educational support | 7 | *The patient is at an age-appropriate grade level but attends resource classes in math and achieves only average grades in other areas.[49]* |
| | | Psychiatric disorder | 3 | *The mother is very focused on the boys' physical and emotional symptoms.[49]* |
| | | Other outcomes only in 1 paper | | Autism: behavioural disturbances: dyslexia: mood disorders |
| Other outcomes | | Other outcomes only in 1 paper | | Schooling: self-identifying as premature |

## CONCLUSION

Parents, patients and clinicians report a wide range of neonatal care outcomes. Parents and patients focus on different outcomes than health professionals. Outcomes reported do not map to organ systems commonly addressed in clinical trials, many are global outcomes. We suggest that the views of former patients and parents should be taken into consideration by researchers and funding bodies.

**Acknowledgements** The authors are grateful to Louise Wann (West Middlesex University Hospital) for her contributions running the database searches.

**Collaborators** COIN Project Steering Group: Elsa Afonso; Iyad Al-Muzaffar; Ginny Brunton; James Duffy; Chris Gale; Anne Greenough; Nigel Hall; Marian Knight; Jos Latour; Neil Marlow; Neena Modi; Laura Noakes; Julie Nycyk; Mehali Patel; Angela Richard-Londt; James Webbe; Ben Wills-Eve.

**Contributors** JW and CG conceived this systematic review. This protocol was created by JW, GB and CG. Searches were performed by LW. All search results were reviewed by JW and assessed by the eligibility criteria. Quality assurance was completed by CG. Coding and result synthesis was completed by JW, GB and

CG. Statistical analysis was completed by NL. The first draft of the manuscript was written by JW, CG, GB and NL. NM edited and reviewed the manuscript. It was approved by JW, CG, GB, SA, LW, NL, NM and the COIN Steering Group.

**Funding** This research is sponsored by Imperial College London and supported by an MRC Clinician Scientist Fellowship award to CG (MR/N008405/1) and salary support for JW from the Portland Hospital.

**Disclaimer** The Imperial College London, the MRC and the Portland Hospital had no involvement in the research or this publication.

**Competing interests** None declared.

**Patient consent** Not required.

**Provenance and peer review** Not commissioned; externally peer reviewed.

**Data sharing statement** Requests for access to data should be addressed to the corresponding author.

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
