## [Reviewer comments · BMJ Paediatrics Open]

This paper was submitted to another journal Archives of Disease in Childhood but declined for publication following peer review. The authors addressed the reviewer's comments and submitted the revised paper to BMJ Paediatrics Open. The paper was subsequently accepted for publication at BMJ Paediatrics Open.

ARTICLE DETAILS

TITLE (PROVISIONAL)	Parent, patient and clinician perceptions of outcomes during and following neonatal care: a systematic review of qualitative research
AUTHORS	Webbe, James; Brunton, Ginny; Ali, Shohaib; Longford, Nicholas; Modi, Neena; Gale, Chris

VERSION 1 - REVIEW

REVIEWER	Reviewer 1 from ADC
REVIEW RETURNED	01-May-2018

GENERAL COMMENTS	The authors report on a very interesting topic, providing a systematic review of neonatal outcomes from the voice of healthcare professionals, ex neonatal patients and parents of patients. I report below my only minor comments: The abstract is well structured. I would suggest to include a more precise description of the findings in the abstract, in order to give a more precise nudge to readers. There is a bias, as far as I can tell, in this manuscript which is evident in the introduction. Are the authors interested in professionals, ex neonatal patients and parents' perceptions of neonatal care or outcomes? The two terms are used interchangeably in the text, but they evidently refer to two different aspects of the early healthcare journey. It was surprising to me not to find Scopus in the included databases. Together with PubMed it has now become one of the largest database for scientific outputs and literature search. Can the authors run a literature search on this database and include additional papers (if any) to their systematic review? As far as I can say, being my research interest on infants and parents admitted to the NICU, some papers from critical authors involved in parents' research in the NICU are missing. Please, check research from these authors: Feeley (e.g., Feeley et al., 2013 – Journal of Clinical Nursing // Journal of Perinatal and Neonatal Nursing); Flacking (e.g., Flacking et al., 2013 – Sexual and Reproductive Health Care); Holditch-Davis (e.g., Holditch-Davis et al., 2015 – Infant Behavior and Development); Jackson (e.g., Jackson et al., 2003 – Journal of Advanced Nursing); Pohlman (e.g., Pohlman, 2005 – Advances in Neonatal Care); Pritchard (e.g., Pritchard & Montgomery-Honger, 2014 – Early Human Development); Provenzi (e.g., Provenzi et al., 2016 – JOGNN // Provenzi & Santoro, 2015 – Journal of Clinical Nursing).
---

	The authors included records published “in a peer review journal in all languages”. How did the authors deal with different languages? Findings are well-presented, in a rich yet well-organized fashion. The discussion are consistent with the methods and the findings. I can say that this manuscript has the potential of being a highly impacting one on research and clinical practice in the neonatal environment. In conclusion, my only two main concerns regard: (1) The precise definition of the focus of this review: is it on neonatal care perceptions? Or is it on outcomes priorities perceptions? (2) Did the authors map the available literature in a comprehensive way? Is the inclusion of Scopus going to fill in the gap of potential relevant papers which are not included in the present form of the manuscript? Are there other potentially relevant papers which have not been included?
--	--

REVIEWER	Reviewer 2 from ADC
REVIEW RETURNED	13-Jun-2018

GENERAL COMMENTS	Thank you for asking me to review this paper which is a comprehensive systematic review of qualitative research examining the opinions of key stakeholders (Parents, staff and ex-patients) on important outcome measures of neonatal care. The systematic review is meticulous and rigorous and the authors should be congratulated on a thorough and detailed review process. The findings are perhaps, not surprising - Healthcare professionals, parents and patients are interested in, or concerned about, different outcomes and importantly parents are particularly concerned about a large number of outcomes that are often quite difficult to measure and standardise such as normality and suffering. Most, if not all, parents want their children to be happy and healthy, to achieve their ambitions and live long and fruitful lives and of course, this is not unreasonable. But these concepts are challenging to measure and standardise in research context as parents expectations and the patients' opportunities offered may vary widely. In this study we are presented with a very long list of outcomes (146) from 62 qualitative studies; 69 of which (almost half) were only mentioned in 1 of the 62 studies. Are these outcomes important? Clearly they are important to some individuals and this should not be ignored, but as they figure so infrequently it probably reflects the unique experience of having a sick baby to individual parents which is the inherent difficulty of interpreting this type of research. Where does this study take us? Clearly parents' views are very important and they should be involved closely in research development (as they are in the UK) but we need to be realistic about what is and is not achievable in terms of consistently and accurately measurable research outcomes. Trying to accommodate the concerns of all runs the risk of 'research waste' the authors warn us about as studies become unwieldy and inordinately expensive.
--

	Perhaps the authors could reflect on this a little more. I would like to see the authors views on how this work takes us forward in a way which will be achievable in the design of future research projects I think the numerous tables with one line quotes and details from individual studies will take up a lot of pages and perhaps some thought could be given to reducing the size and detail without losing the overall message.
--	---

VERSION 1 – AUTHOR RESPONSE

REVIEWER COMMENTS:

Reviewer #1

Reviewer comment:

There is a bias, as far as I can tell, in this manuscript which is evident in the introduction. Are the authors interested in professionals, ex neonatal patients and parents' perceptions of neonatal care or outcomes? The two terms are used interchangeably in the text, but they evidently refer to two different aspects of the early healthcare journey.

Response:

This manuscript is focussed on outcomes. There are two sources of confusion: how 'perceptions of care' are handled in this manuscript and the use of qualitative data.

Firstly, as highlighted by the reviewer this manuscript does contain text relating to 'perceptions of care', but only because this is one of many possible outcomes. These outcomes do include outcomes relating to how neonatal illness and care affect the lives of parents, staff and the wider society. We hope that this work supports the growing body of evidence that neonatal illnesses has consequences and effects that extend beyond the sick baby.

Secondly, using qualitative data in an evidence synthesis does necessitate relying on the perceptions of stakeholders (as all qualitative data includes the perception of study participants). We have limited our focus to how professionals, ex-patients and parents perceive the outcomes of neonatal care.

To reinforce the focus on outcomes we have included the COMET definition of outcomes in the introduction (Page 5, Paragraph 2).

"An outcome is then measured effect that illness or treatment has on an individual."

Reviewer comment:

It was surprising to me not to find Scopus in the included databases. Together with PubMed it has now become one of the largest database for scientific outputs and literature search.

Response:

There are, of course, a number of databases that could be used to undertake this systematic review. We are confident that by including CINAHL, PsycInfo, EMBASE and ASSIA in addition to MEDLINE we have covered a broad range of literature (including social sciences and nursing research). With hindsight it might have been easier just to search Scopus – but we are confident that the extended range of databases we have searched is a strength of our paper.

Reviewer comment:

As far as I can say, being my research interest on infants and parents admitted to the NICU, some papers from critical authors involved in parents' research in the NICU are missing. Please, check research from these authors: Feeley (e.g., Feeley et al., 2013 – Journal of Clinical Nursing // Journal of Perinatal and Neonatal Nursing); Flacking (e.g., Flacking et al., 2013 – Sexual and Reproductive Health Care); Holditch-Davis (e.g., Holditch-Davis et al., 2015 – Infant Behavior and Development); Jackson (e.g., Jackson et al., 2003 – Journal of Advanced Nursing); Pohlman (e.g., Pohlman, 2005 – Advances in Neonatal Care); Pritchard (e.g., Pritchard & Montgomery-Honger, 2014 – Early Human Development); Provenzi (e.g., Provenzi et al., 2016 – JOGNN // Provenzi & Santoro, 2015 – Journal of Clinical Nursing)..

Response:

The reviewer highlights important papers by notable authors in the field. In several cases we have included other papers by these authors. We have reviewed the papers suggested and can confirm that these papers have not been included in our review because they do not discuss outcomes. In many cases the papers do not contain the word outcome at all. As discussed above while there is some overlap between experiences of neonatal care and the outcomes of care our review is specifically focussed on outcomes and as such these papers do not fulfil our inclusion criteria.

Reviewer comment:

The authors included records published “in a peer review journal in all languages”. How did the authors deal with different languages?

Response:

In most cases a copy of the paper was available in English. In a small number of cases the paper was only available in a different language (specifically Portuguese) a translation was obtained.

Reviewer #2

Reviewer comment:

In this study we are presented with a very long list of outcomes (146) from 62 qualitative studies; 69 of which (almost half) were only mentioned in 1 of the 62 studies. Are these outcomes important? Clearly they are important to some individuals and this should not be ignored, but as they figure so infrequently it probably reflects the unique experience of having a sick baby to individual parents which is the inherent difficulty of interpreting this type of research.

Response:

We agree that a limitation of this work is that we are not able to attach a measure of ‘importance’ to different outcomes, we can only discuss how frequently the outcome is discussed. We have amended the tables (as described above) to make them easier to comprehend without losing information on these outcomes completely.

Reviewer comment:

Clearly parents' views are very important and they should be involved closely in research development (as they are in the UK) but we need to be realistic about what is and is not achievable in terms of consistently and accurately measurable research outcomes. Trying to accommodate the concerns of all runs the risk of 'research waste' the authors warn us about as studies become unwieldy and inordinately expensive.

Perhaps the authors could reflect on this a little more. I would like to see the authors views on how this work takes us forward in a way which will be achievable in the design of future research projects

Response:

We agree that parents' views are very important and hope that this work highlights how different parents' views can be. As a practicing clinician I was surprised by how far what parents and patients in particular seemed to be discussing in these papers was from what I expected. We hope that this review will flag these differences to your readers and perhaps direct them towards some of the original papers.

We strongly believe that involving all parties in a constructive discussion about outcome selection will help to reduce research waste. The 'Core Outcomes In Neonatology' project that we mention in this paper is an attempt to undertake this in a rigorous manner using standardised methodology. We agree with the reviewer that it will be important that this work avoids the pitfall of recommending numerous impractical outcome measures and tools. This has to be a true collaboration between groups, and we will be discussing these issues when we publish the results of our core outcome set development work in the future.

Reviewer comment:

I think the numerous tables with one line quotes and details from individual studies will take up a lot of pages and perhaps some thought could be given to reducing the size and detail without losing the overall message.

Response:

The tables have been amended as described above.

VERSION 2 – REVIEW

REVIEWER	Reviewer 1 from ADC
REVIEW RETURNED	14-Aug-2018

GENERAL COMMENTS	Thanks for the careful and detailed replies to my concerns and suggestions. I see no more issues in your manuscript.
---